# The Effects of Caffeine on Blood Platelets and the Cardiovascular System through Adenosine Receptors

**DOI:** 10.3390/ijms25168905

**Published:** 2024-08-15

**Authors:** Kinga Marcinek, Boguslawa Luzak, Marcin Rozalski

**Affiliations:** Department of Haemostasis and Haemostatic Disorders, Chair of Biomedical Sciences, Faculty of Health Sciences, Medical University of Lodz, Mazowiecka 6/8, 92-235 Lodz, Poland; kinga.bartczak2@stud.umed.lodz.pl (K.M.); boguslawa.luzak@umed.lodz.pl (B.L.)

**Keywords:** adenosine receptors, antiplatelet therapy, caffeine, cardiovascular system

## Abstract

Caffeine is the most popular and widely consumed behaviourally active substance in the world. This review describes the influence of caffeine on the cardiovascular system, with a special focus on blood platelets. For many years, caffeine was thought to have a negative effect on the cardiovascular system mainly due to increasing blood pressure. However, more recent data suggest that habitual caffeine consumption may reduce the risk of cardiovascular disease and hypertension. This could be a significant finding as cardiovascular disease is the leading cause of death worldwide. Caffeine is known to inhibit A_1_ adenosine receptors, through which it is believed to modulate inter alia coronary blood flow, total peripheral resistance, diuresis, and heart rate. It has been shown that coffee possesses antiplatelet activity, but depending on the dose and the term of its use, caffeine may stimulate or inhibit platelet reactivity. Also, chronic exposure to caffeine may sensitize or upregulate the adenosine receptors in platelets causing increased cAMP accumulation and anti-aggregatory effects and decrease calcium levels elicited by AR agonists. The search for new, selective, and safe AR agonists is one of the new strategies for improving antiplatelet therapy involving targeting multiple pathways of platelet activation. Therefore, this review examines the AR-dependent impact of caffeine on blood platelets in the presence of adenosine receptor agonists.

## 1. Introduction

Caffeine (1,3,7 trimethylxanthine) is one of the most frequently consumed behaviourally active substances in the world. It is commonly found in coffee, tea, various soft and energy drinks, and chocolate [1]. The richest source of caffeine is coffee, one of the most popular beverages in the world [2,3,4]. Global coffee consumption in 2021/2022 is estimated to be 175.6 million bags, with a bag containing 60 kg of coffee [5]. Drinking coffee has become a daily habit for many people, with Americans consuming around 517 million cups of coffee every day [6], the mean daily consumption per capita being two to three cups of coffee [6].

The cardiovascular system is composed of the heart, blood vessels, and blood, with the heart arguably being the most important. The blood vessels form a network of arteries, veins and capillaries, whose main function is to carry blood. The primary role of blood is of course to carry oxygen and other nutrients to various organs and carbon dioxide to the lungs for removal.

Cardiovascular disease (CVD), a term referring to all diseases that affect the circulatory system, including the heart and blood vessels, is known to be the leading cause of death worldwide [7], with more than 500 million people suffering from heart and coronary disease [8]. More importantly, this trend is increasing [9], and 20.5 million deaths were caused by cardiovascular disease in 2021 [8]. Several socioeconomic, metabolic, behavioural, and environmental risk factors have been identified for CVD, including hypertension, tobacco use, excessive alcohol consumption, unhealthy diet, physical inactivity, diabetes and obesity together with the effects of air pollution, kidney disease, and stress [8,9]. However, although the number of deaths from CVD has increased worldwide, the age-standardised death rate appears to have fallen by a one-third over the last three decades [8].

There is no doubt that caffeine affects the human organism. The structure of caffeine molecule, as a heterocyclic organic compound with a purine base, is very similar to the structures of other metabolic compounds, including adenosine. Caffeine is believed to exert its influence via the activity of adenosine receptors (ARs), ryanodine receptors (RyRs), and gamma-aminobutyric acid receptors (GABA receptors) or by inhibiting phosphodiesterase [10]. For many years, caffeine consumption has been believed to have a negative effect on the cardiovascular system [11]; however, recent data suggest that coffee consumption may in fact have several positive effects on human health [12,13].

It has been proposed that caffeine exerts most of its biological effects via adenosine receptors [14]. Also, through ARs: A_2A_R and A_2B_R, caffeine may modulate platelet function, the effect of which may be an intensification or weakening of platelet reactivity depending on the dose and frequency of consumption. Caffeine has been shown to alter the sensitivity of ARs to adenosine or other AR agonists, but there are few studies on this topic. As such, the aim of the present review is to examine the effects of caffeine on blood platelets and the cardiovascular system via ARs. It is also assumed that coffee is the richest source of caffeine and most popular caffeinated beverage in the world. The interaction between caffeine and the cardiovascular system is of particular importance considering the high mortality associated with CVD, and our findings may indicate areas for possible future research and searching the new strategies to improve antiplatelet therapy involving targeting multiple pathways of platelet activation.

In this review, we have used PubMed, Scopus, and Web of Science as the major databases for searching the scientific literature published prior to 31 March 2024. Additionally, we have used Google Scholar for retrieving more general data, such as reports. We used the following keywords (or their synonyms) and their combinations: platelet(s), caffeine, adenosine, adenosine receptor(s), AR, coffee, cardiovascular, endothelium, and antiplatelet therapy.

## 2. Characteristics of Caffeine

### 2.1. General Information

Caffeine (1,3,7-trimethylxanthine) is a purine alkaloid and the most widely consumed behavioural stimulant across the globe [10]. In its pure state, it exists as a white, odourless, fat- and water-soluble powder [10,15]. It is known to occur naturally in over sixty plant species, although it is primarily found in coffee beans, tea leaves, and cocoa beans, as well as guarana berries and kola nuts [10,15,16]. Caffeine itself was first discovered in tea (*Camellia sinensis* L.) and coffee (*Coffea arabica* L.) in 1820 [17]. Currently, the richest and most popular natural source of caffeine is believed to be coffee. It is obtained from the roasted beans of *Coffea arabica* L. (Arabica coffee) and *Coffea canephora* Pierre ex Froehner (Robusta coffee), both of which belong to the *Rubiaceae* family.

### 2.2. Metabolism of Caffeine in Humans

In humans, almost 99% of caffeine is absorbed within 45 min of ingestion, mostly in the gastrointestinal tract [10,18]. The absorption process may be affected by various factors, including pH or route of administration [19]. Following oral intake, the maximum plasma concentration is achieved at 15–120 min [10,15,18]. It typically takes 45–60 min for caffeine to produce its effect, although subsequent food intake may delay this [10,15]. Caffeine has a metabolic half-life of approximately three to five hours and can cross the blood–brain barrier; it can also enter various bodily fluids including serum, milk, saliva, and semen [10,15,19,20].

Caffeine metabolism in humans includes multiple and separate pathways with demethylation to dimethylxanthines and monomethylxanthines, C8 oxidation of these methylxanthines into methylurates, and ring opening yielding substituted uracil derivatives [21]. Most caffeine is metabolised in the liver (99%), and this process is mediated by the cytochrome P450 oxidase enzyme system [10,22]. In the first step, the caffeine is subjected to 3-demethylation by cytochrome P450 isoform 1A2 (CYP1A2), leading to the formation of its three main primary metabolites: paraxanthine (PX, 1,7-methylxanthine), theobromine (TB, 3,7-methylxanthine), and theophylline (TP, 1,3-methylxanthine) in the proportion of 80:12:7, and to a lower extent to trimethylated urate (1,3,7-methyluric acid (1,3,7-MU) [23]. The main caffeine metabolites (paraxanthine, theophylline, and theobromine) have similar biological activity to caffeine. A small proportion of the ingested caffeine (0.5–4.0%) is excreted in urine and bile without undergoing any change [20,24]. Paraxanthine is the major caffeine metabolite and should be taken into consideration when studying caffeine effects on the body. After caffeine intake, its plasma concentrations decrease more rapidly than those of its metabolite paraxanthine. The half-lives of theophylline and theobromine were significantly longer than those of caffeine and paraxanthine [21]. Further metabolism, which includes the transformation (further oxidation and demethylation) of paraxantine, theophiline, and theobromine, results in secondary metabolites such as monomethylxanthines (1-MX,1-methylxanthine; 3-MX, 3-methylxanthine; and 7-MX, 7-methylxanthine), dimethylurate derivates (1,3-MU, 1,3-methyluric acid and 1,7-MU, 1,7-methyluric acid), and monomethylurates (1-MU, 1-methyluric acid) [23]. These secondary metabolites are excreted in the urine.

Caffeine metabolism is known to be influenced by individual variations in CYP1A2 activity [25], which can range from 5- to 15-fold in healthy individuals in response to various environmental and genetic factors [26]. A key factor believed to induce CYP1A2 activity is the consumption of caffeinated products, particularly coffee [24,26]. Additionally, various lifestyles have been linked to an increase in CYP1A2 activity; for example, higher coffee consumption has been noted among smokers [27], and the use of oral contraceptives, pregnancy, obesity, or alcohol consumption, has been found to affect caffeine metabolism [24].

Genetic factors also play an important role in caffeine metabolism [25], with two genes being associated with this process: *CYP1A2* and *ADORA2A* [10]. A single nucleotide polymorphism (SNP) within *CYP1A2* (-163C > A; rs762551) has been reported to exert an impact on CYP1A2 induction [10,28]; it appears to differentiate between “fast” (AA genotype) and “slow” (AC or CC genotype) caffeine metabolizers [10,29]. The rs5751876 SNP in the *ADORA2A* gene [30] also influences the response to caffeine, with the TT genotype indicating “high responders to caffeine” and the CC/CT genotype indicating “low responders to caffeine” [29]. These terms denote individual sensitivity to caffeine and thus variability in caffeine metabolism [10]. The metabolism of caffeine is presented in the Figure 1.

### 2.3. Biological Effects of Caffeine

Caffeine affects the cardiovascular, nervous, immune, respiratory, renal, digestive, and musculoskeletal systems [10,18,31]. Both caffeine and its metabolites have the ability to modify lipid and glucose metabolism [32,33]. Caffeine is known to affect the sympathetic nervous system resulting in the secretion of epinephrine and norepinephrine [10].

Caffeine also stimulates the central nervous system and enhances hormonal secretion [34]. It elevates neurotransmitter levels in the brain, resulting in behavioural and neurochemical changes [34] and has been shown to possess neuroprotective properties [19]. Moreover, caffeine plays a crucial role in inflammatory processes in a dose-dependent manner [35]. It appears that minimal caffeine levels could have negative effects, while higher concentrations may reduce inflammatory biomarkers and also activate anti-inflammatory mechanisms [10,36]. Furthermore, experimental studies suggest that caffeine may play a role in the antioxidant properties of coffee [34].

Caffeine can influence cells by blocking adenosine receptors, releasing stored intracellular calcium and inhibiting phosphodiesterases (PDEs) [18]. It can block all adenosine receptors, in particular, A_1_R and A_2A_R [10,16]. It is worth noting that while caffeine can block adenosine receptors at low doses, higher levels are needed for the two other mechanisms [16]. Caffeine can also induce calcium release from the endoplasmic reticulum [10] by binding to ryanodine receptors (RyRs) [37]; this also inhibits calcium reuptake [38]. As a result of these processes, caffeine can enhance contractility during submaximal contractions in both habitual and non-habitual caffeine consumers [16]. Finally, it is also a non-selective competitive inhibitor of phosphodiesterases, which can increase cellular cAMP concentration [10]. The accumulation of cAMP stimulates lipolysis and the release of various hormones, such as dopamine, epinephrine, and norepinephrine [16].

### 2.4. Adverse Effects and Toxicity of Caffeine

The effect of caffeine depends mainly on the amount of its intake. Moderate caffeine consumption, i.e., up to 400 mg per day, is generally considered to be safe for adults [10,31,39]. Moreover, a single dose of caffeine up to 200 mg poses no safety concerns for healthy adults [39]. Nonetheless, high single doses of caffeine above 300 mg may result in acute caffeine intoxication [10]. The most frequent symptoms are nervousness, restlessness, irritability, anxiety, or insomnia [40], as well as manifestations related to the cardiovascular, gastrointestinal, and urinary systems [10]. Increased caffeine consumption can induce anxiety and affect mood [41]. A chronic intake of caffeine higher than 400–600 mg/day has been associated with diverse dysfunctions [31]. Intake of 100 mg or more of caffeine may prolong sleep onset and decrease sleep duration, particularly when consumed close to bedtime [31,39]. The majority of negative side effects are related to the stimulatory effects of caffeine [31].

The negative effects of caffeine, like its positive influence on the human organism, are the result of the caffeine interaction with receptors such as ARs, RyRs, and GABA receptors or inhibition of PDEs. The circulatory system is one of the most vulnerable to the negative effects of caffeine because of its impact on blood pressure (both systolic and diastolic). The stimulating action of caffeine may be associated with an increase in intracellular calcium concentrations, the release of norepinephrine, and the sensitization of dopamine receptors through inhibition of ARs and PDEs and activation of the β1-receptor, which can lead to problems related to cardiac function such as tachycardia and arrhythmia. Caffeine, as a nonspecific inhibitor of PDEs, is able to intensify the production of cAMP and cGMP, resulting in affecting cardiac contractility, and this may predispose to arrhythmias [10]. In the brain, caffeine has multiplate targets, and there are also ARs, RyRs, and GABA receptors or PDEs. Its action on A_2A_Rs may explain the psychomotor stimulant effect, mediated by dopaminergic mechanisms. Caffeine, through antagonism of ARs, affects brain functions such as sleep, cognition, learning, and memory and modifies brain dysfunctions and diseases: Alzheimer’s disease, Parkinson’s disease, Huntington’s disease, epilepsy, pain/migraine, depression, and schizophrenia [10].

Furthermore, caffeine appears to have addictive properties. Sudden discontinuation of regular caffeine intake can lead to caffeine withdrawal syndrome, as described in the Diagnostic and Statistical Manual of Mental Disorders of the American Psychiatric Association (APA) [42]. Symptoms include headache, drowsiness, lethargy, fatigue, reduced motivation, self-confidence, and increased irritability, as well as a decrease in blood pressure [31,43]. Caffeine withdrawal symptoms occur 12–24 h after the last consumption of caffeine and reach their peak after 20–48 h [31]. To prevent caffeine withdrawal syndrome, gradual reduction in ingested caffeine doses is recommended.

For many years, caffeine consumption has been believed to be associated with negative health outcomes, particularly related to cardiovascular events. Recently, however, this opinion has been subjected to a significant re-evaluation. The impact of caffeine on the cardiovascular system is therefore reviewed in the following paragraphs and the summary of its biological effects is presented in the Table 1.

### 2.5. Caffeine Interaction with Drugs

Simultaneous consumption of caffeine and taking certain medications can lead to changes in the therapeutic effects of these drugs. Taking caffeine with stimulants like amphetamines, cocaine, ephedrine, or medications for ADHD can increase side effects like jitteriness, rapid heartbeat, and anxiety. Caffeine decreases the effectiveness at lowering blood pressure of the blood pressure medications like beta blockers and diuretics. Additionally, caffeine can decrease the sedative effects of benzodiazepines used for anxiety or sleep making them less effective [44]. Besides, caution is required to caffeine consumption during antibiotic treatment.

Recently, special attention has been paid to the impact of caffeine on the effects of drugs used during the non-invasive and invasive functional measurements used to assess myocardial perfusion when maximal hyperaemia is essential. The most widely used vasodilator agents used to achieve this hyperaemic effect are adenosine, regadenoson, and dipyridamole. The hyperaemic effect is primarily caused by binding to the adenosine A_2A_-receptor on arteriolar vascular smooth muscle cells. Regadenoson can safely be used in patients with hypersensitive airways, due to its selective binding to the A_2A_-receptor. It was reported that the use of regadenoson stress for myocardial perfusion imaging in caffeine consumers is very common, safe, and associated with a lower incidence of certain symptoms than in non-caffeine consumers [45]. The available data indicate a significant influence of recent caffeine intake on cardiac perfusion measurements during adenosine- and dipyridamole-induced hyperaemia [46].

## 3. Adenosine Receptor Characteristics

### 3.1. Overview

Adenosine plays an important role in the human body. It exerts its effects through the activation of four different adenosine receptors (ARs): A_1_, A_2A_, A_2B_, and A_3_ [47,48]. Each AR has the same core domain but is characterised by a different number of amino acids [49]. Adenosine receptors are G protein-coupled receptors (purinergic receptors, P1R), classified as either adenylyl cyclase inhibitors (A_1_ and A_3_) or adenylyl cyclase activators (A_2A_ and A_2B_) [50]. While stimulation of A_2A_ and A_2B_ increases cAMP production, followed by the activation of protein kinase A (PKA) and phosphorylation of CREB protein (cAMP response element-binding protein), stimulation of A_1_ and A_3_AR has the opposite effect [51].

Although G protein-coupled receptors (GPCRs) were previously viewed as monomeric units interacting one-to-one with their respective heterotrimeric G proteins, this view has changed [52]. It has been shown that GPCRs (including adenosine receptors) can occur in homomeric, oligomeric, and heteromeric forms and that heteromers are characterised by novel functional (signalling) properties compared to homomers [53,54]. A_2A_R heteromers can be found in several combinations, including A_1_R-A_2A_R, A_2A_R-A_2B_R, A_2A_R-D2R, A_2A_R-D3R, and A_2A_R-D4R4; however, the A_2A_R-D2R heteroreceptor complex is of the greatest interest as it plays a role in Parkinson’s disease, schizophrenia, and cocaine addiction [53].

### 3.2. Function of Adenosine Receptors

Adenosine receptors are widely expressed and have been implicated in several physiological and pathological biological functions. They are found in among others, the neurological, cardiovascular, respiratory, gastrointestinal, urogenital, and immunological systems [49,55]. In addition, ARs can also be found in the bones, joints, eyes, and skin [49,55]. Each AR has a distinct cell and tissue distribution and secondary signalling transducers [49]. The A_1_AR and A_3_AR signals are mediated by G_i_ and G_o_ proteins, while the other two are linked to G_o_ proteins [47]. However, all AR subtypes are present in the heart [56].

A_1_AR is present in the atrium, smooth muscles, and endothelial coronary tissues [57,58]. It can be also found in the central nervous system (CNS) with high levels in the cerebral cortex, hippocampus, cerebellum, thalamus, brainstem, and spinal cord [59]. It regulates functions such as neurotransmitter release, reduction in neuronal excitability, or pain relief [49,60]. However, it is also responsible for some negative effects in other tissues and systems: it reduces renal blood flow and suppresses insulin secretion and lipolysis [49,51] and is responsible for pro-inflammatory effects in several immune cells [48,49].

The A_3_AR subtype can be found in a variety of primary cells, tissues, and cell lines [49,59]. It is expressed in smooth muscle cells such as coronary artery muscle cells [61,62]. It is also able to mediate anti-inflammatory effects in inflammatory cells such as eosinophils, neutrophils, monocytes, and macrophages [49]. The A_3_AR subtype has also been implicated in allergic responses and apoptosis [59].

The A_2A_AR subtype is widely expressed centrally and peripherally but mostly in the striatum, olfactory tubercle, and immune system [49]. This subtype can be found in leukocytes, platelets, and blood vessels, where its role is to mediate a variety of anti-inflammatory, anti-aggregatory, and vasodilatory effects [59,60]. In the cardiovascular system, A_2A_AR is expressed in the vessels, atria, and ventricular tissues [57,63]. It also plays an important role in Parkinson’s disease, Huntington’s disease, Alzheimer’s disease, and ischaemia [59]. Interestingly, both the A_3_AR and A_2A_AR subtypes are associated with cancer, but they have different effects [49]. In addition, activation of the A_2A_AR pathway suppresses the synthesis and release of inflammatory cytokines [64].

Finally, the A_2B_AR subtype can be found in various cell types in the intestine, bladder, vas deferens, and lungs [59]. It has also been observed on myocytes and fibroblasts and in the smooth muscles of coronary arteries [65,66]. Like the other subtypes, A_2B_AR may also play a role in modulating inflammation and immune processes [67]. This subtype appears to be upregulated in hypoxia, inflammation, and cell stress [49].

The most important mechanism of action of methylxanthines, including caffeine and its primary metabolites, involves blocking the adenosine receptors and competitively inhibiting the action of adenosine in the cells. Both caffeine and theophylline are potent inhibitors of adenosine receptors in the human brain. However, theophylline and paraxanthine were proposed to have slightly higher affinities than caffeine for the adenosine A_1_, A_2A_, and A_2B_ receptors and to also be weak antagonists for the adenosine A_3_ receptor subtype. The inhibitory potency of methylxanthines is much more diverse in the receptors of the A_2_ subtype (A_2A_ plus A_2B_). IC_50_ values for effective blockade of receptors of the A_2_ subtype were 45 and 98 µM for theophylline and caffeine, respectively, and 2500 µM for theobromine. Theobromine (not possessing the 1-methyl group reported as important for adenosine antagonism) was in fact reported to have significantly lower affinity than caffeine for A_1_ and A_2A_ receptor subtypes [68]. The role of adenosine receptors in the cardiovascular system is summarised in the Table 2.

## 4. Role of Platelets in Cardiovascular System Function

Among the various roles of platelets, one of their most important is their involvement in haemostasis and thrombosis. In pathological conditions, platelets are activated to prevent excessive blood loss. During primary haemostasis, activated cells adhere to the vessel endothelium and form aggregates. Additionally, platelets play a significant part in thrombosis. Upon their activation, platelets release several substances including fibrinogen, which is then converted into fibrin; this supports their role in thrombosis and in wound healing [75]. Platelets also release various substances to support wound healing, such as growth factors, cytokines, and chemokines [76] and enhance thrombin production [76]. However, while their involvement in both haemostasis and thrombosis is widely acknowledged, their other roles should not be disregarded.

Platelets play a crucial role in maintaining vascular integrity and proper blood flow within vessels [77,78]. Studies have shown that platelets support the vascular endothelium through four mechanisms: filling potential gaps, stimulating endothelial cell growth, assisting in the maintenance of endothelium ultrastructure, and finally releasing factors that improve endothelial barrier function [78]. They also exhibit proangiogenic properties through the use of the pro- and anti-angiogenic factors within their granules [78]. Platelets also participate in inflammatory processes and thus possess a significant role in inflammation [75].

However, while platelets appear to contribute to the development of cardiovascular disease, they may also provide cardioprotective benefits. They have also been shown to support heart muscle regeneration after a myocardial infarction (MI) [79]. Furthermore, their protective properties are evident in their ability to preserve vascular integrity in the face of inflammation [78].

## 5. Adenosine Receptor-Mediated Effects of Caffeine on Platelets

Human blood platelets express receptors A_2A_AR and A_2B_AR [80,81]. Platelets have a significantly greater density of A_2A_ARs than A_2B_ARs, and previous studies suggest A_2A_AR is the sole receptor involved in adenosine-stimulated platelet activation [82]. Early studies identified A_2A_AR to be an important receptor expressed in platelets and to mediate adenosine inhibition of platelet aggregation. Caffeine is proposed to act as a weak A_2A_R antagonist; however, some functional studies based on cells and experimental Parkinson’s suggest that it may behave as an inverse agonist [83].

Significantly fewer studies have investigated the effects of coffee on platelet function compared to those examining the effect on serum lipids and blood pressure [84,85,86,87,88,89,90,91,92,93]. Generally, it seems reasonable to conclude that coffee extracts are effective in inhibiting platelet aggregation, a critical step of primary haemostasis and thrombosis. It has been shown that coffee possesses antiplatelet activity and inhibits platelet aggregation induced by adenosine diphosphate, collagen, arachidonic acid, and epinephrine [85,86,90,92,93]. In another study, coffee extracts were found to have anti-aggregatory effects on in vitro platelet aggregation induced by adenosine diphosphate (ADP) or arachidonate but not by collagen [87]. Some authors suggested that the antiplatelet effect of coffee is probably independent from caffeine and could be a result of the interaction of coffee phenolic acids with the intracellular signalling network, which is known to lead to platelet aggregation [93].

Nevertheless, it has also been observed that caffeine might stimulate platelet reactivity in vivo; the administration of 100 mg of caffeine has been found to increase the release of *β* thromboglobulin one hour later [84]. Cavalcante et al. report no changes in ADP-, collagen- and adrenaline-stimulated platelet aggregation in individuals receiving 750 mg/day of caffeine; although diastolic pressure significantly increased 24 h after the first intake, this effect disappeared in the following seven days [89].

In other in vitro studies, samples treated with caffeine (770 µM or 1540 µM) demonstrated significantly lower mean platelet aggregation following epinephrine or ADP treatment compared to control samples. No significant differences were observed in mean values of collagen- or ristocetin-induced platelet aggregability. These results suggest that caffeine selectively inhibits epinephrine- and ADP-stimulated platelet aggregation and disturbs the release of endogenous ADP from platelets in response to exogenous ADP [94].

It was also observed that chronic caffeine intake could affect ADP-induced platelet aggregation but only when the intake lasted longer than one week and was higher than 400 mg/d [93]. Coffee consumption was found to significantly reduce collagen- and arachidonic acid-stimulated blood platelet aggregation at both the 30- and 60-min time points; however, coffee had no significant influence in blood platelet aggregation stimulated by ADP [93]. Hence, coffee consumption was found to have antiplatelet effects.

It has been proposed that caffeine exerts its antiplatelet activity via 3′,5′-cyclic adenosine monophosphate (cAMP), an important second messenger in blood platelets. It was found that caffeine increased the level of cAMP, most likely by inhibition of phosphodiesterase. In vivo studies indicate that coffee consumption significantly inhibited phosphodiesterase activity and selected coffee constituents such as caffeine, theophylline, paraxanthine, and caffeine metabolites (0.1–5 mM), inhibited PDEs activity in blood platelets in vitro. The authors suggest that moderate consumption of coffee may modulate blood platelet aggregation, at least in part by altering PDEs activity and cAMP homeostasis [95].

Baggioni et al. found that chronic intake of coffee alters the response of platelets to the activity of adenosine- or NECA-induced (NECA—non-selective agonist of adenosine receptor) platelet inhibition [1]; the authors propose that chronic exposure to caffeine may sensitize or upregulate the adenosine receptors. Similarly, [96] observed that a seven-day administration of 750 mg caffeine per day (3 × 250 mg) resulted in changes in ADP-induced platelet aggregation and increased cAMP accumulation and A_2A_ adenosine receptor upregulation 12 or 60 h after the last dose of caffeine [96]. In addition, Varani et al. report that the intake of caffeine (400 or 600 mg/day for one week or 400 mg/d for two weeks) may upregulate A_2A_ adenosine receptor activity, increase cAMP accumulation and anti-aggregatory effects, and decrease calcium levels elicited by HE-NECA in a time-dependent and dose-dependent manner [97].

Although such changes can influence platelet function, caffeine tolerance probably does not alter the potency of caffeine as a competitive antagonist to adenosine. It is more likely that caffeine tolerance may be increased by other changes, such as a shift in receptors to a high-affinity state, alterations in G protein levels or the coupling of these proteins to adenosine receptors, or long-term receptor occupancy. It is nevertheless possible that while chronic caffeine intake may enhance the antiplatelet activity of adenosine and its analogues, acute intake may lower the activity due to the antagonism of A_2A_ receptors.

### Can Caffeine Modulate Effects of Inhibitors of Adenosine Receptors on Platelets?

The activation of blood platelets plays a crucial role in the initiation and development of arterial thrombotic diseases, such as coronary heart disease and stroke. Therefore, the ‘first choice’ in the management and treatment of arterial thrombotic disorders is antiplatelet therapy. Despite the fact that many antiplatelet agents are currently available and that some of them are approved and widely applied, an effective treatment of arterial thrombosis remains elusive. This problem is twofold: either antiplatelet agents (e.g., aspirin) interfere with only one of the several pathways of platelet activation or they may (e.g., fibrinogen receptor blockers) very effectively block the final common step but pose a risk of bleeding. Also, the activity of many antiplatelet agents varies considerably in response to environmental or genetic factors. As such, still there is a need for novel platelet inhibitors with better efficacy and safety or treatments based on a combined therapy of currently available agents.

One promising strategy for effective antiplatelet therapy is based on targeting multiple platelet activation pathways. Recently, it was proposed that the inhibition of platelet function by P2Y_12_ antagonists can be potentiated by adenosine receptor agonists [74]. In addition, exposure to certain adenosine receptor agonists—UK432097, 2-chloroadenosine, MRE0094, and PSB0777—preserved platelet viability with no cytotoxic effects being observed [98]. A group of AR agonists was found to have significant antiplatelet activity, as indicated by multiple parameters of platelet function: aggregation in whole blood and platelet-rich plasma [98], P-selectin expression, GPIIb-IIIa activation, fibrinogen binding, and calcium ion mobilization [99]. The results confirm that AR agonists demonstrate a synergistic effect with P2Y_12_ antagonists, resulting in greater inhibition of ADP-induced platelet activation; in addition, it was found that the AR agonists, both alone and in combination with a P2Y_12_ antagonist, increase cAMP formation in both unstimulated and ADP-activated platelets, suggesting that the mechanism is cAMP-dependent [99].

Interestingly, the antiplatelet effects of AR agonists were found to be heterogeneous in a combined model comprising blood samples incubated with AR agonists and cangrelor or prasugrel metabolites (PMs). When a whole group of healthy subjects was divided into two subpopulations—high responders and low responders—to P2Y_12_ receptor inhibitors, separately for cangrelor and PM, a statistically significant difference was found between both responder groups for each agonist–antagonist pairing [100]. The P2Y_12_ antagonist low responders demonstrated markedly higher inhibition of AR-stimulated platelet response than high responders, as indicated by the aggregation increase factor, i.e., the number of times an anti-aggregatory effect is intensified by an AR agonist. Interestingly, combinations of stronger AR agonists (NECA and to a smaller extent regadenoson) with a P2Y_12_ antagonist achieved comparable inhibition of overall aggregation in both the high- and low-responder groups [100]. This suggests that patients with resistance to P2Y_12_ inhibitors could potentially particularly benefit from adenosine receptor-targeted therapy.

The effect of AR agonists and P2Y_12_ inhibitors on thrombus formation was also assayed with the use of the Venaflux system using Vena8 Fluoro+ biochips coated with type I collagen. Blood samples incubated with AR agonists (NECA 0.5 µM, regadenoson 1.2 µM) and/or P2Y_12_ inhibitor cangrelor (17 nM) were perfused through the channels of a chip using a shear force of 60 dyn/cm^2^. Under applied conditions, the AR agonists (NECA and regadenoson) alone decreased clot formation by a mean value of 84%. Cangrelor alone decreased clot formation by 68.3%. Interestingly, pairing the AR agonists with a P2Y_12_ inhibitor improved the antiplatelet effect. In general, the observed inhibitory effect on thrombus formation was stronger than could be expected, taking into account the results obtained in aggregation experiments [99].

Finally, these effects were confirmed using an in vivo model of ferric chloride-induced thrombosis in mice. In this model, HE-NECA was found to potentiate the anti-thrombotic effects of either cangrelor or prasugrel. HE-NECA demonstrated efficacy both when administered chronically and in bolus form, as noted previously for other AR agonists [101]. It is known that AR agonists can elevate blood–brain barrier permeability and have a hypotensive impact. Indeed, it was found that bolus HE-NECA administration significantly reduced blood pressure, although this impact was much less pronounced when the drug was administered in dosages equivalent to those achieved from chronic administration. In addition, HE-NECA was not observed to have any significant effect on the blood–brain barrier [101].

Generally, these findings support the idea of dual antiplatelet therapy; they confirm the potential value of platelet adenosine receptors as therapeutic targets and the benefits of therapeutic use of combined P2Y_12_ antagonist and AR agonist therapy in cardioprotection.

In the light of above findings, it would be interesting to verify the role of caffeine and its metabolites in the reduction in platelet reactivity by selective adenosine agonists of A_2A_ receptors. Not only caffeine but also their main metabolites may modulate cell function via adenosine receptors and inhibition of cyclic nucleotide phosphodiesterases. Interestingly, chronic abuse of caffeine in humans (drinking coffee) leads to the overexpression of A_2A_R in platelets accompanied by platelet sensitization and stronger reaction to adenosine analogue HE-NECA and reduction in platelet activation. In the context of antiplatelet activity of caffeine per se or in combination with adenosine analogues, the mechanism related to the increasing in intraplatelet cAMP concentration via PDE inhibition is the most probable, but it is also possible that the analogues with the stronger affinity for the A_2A_ receptor could be less influenced by caffeine than adenosine. Additionally, the effects of the direct interaction of caffeine and its metabolites with A_2A_ receptors in the context of the adenosine analogues’ influence on platelets are less known. It would also be interesting to investigate whether the antiplatelet activity of A_2A_R agonists differs between low and fast caffeine metabolisers or is modulated by nucleotide polymorphisms in A_2A_R. Therefore, it cannot be excluded that the metabolism of caffeine largely determines the variability of the antiplatelet activity of the selected adenosine A_2A_ receptor agonists. Currently, however, according to our knowledge, there is no direct experimental evidence supporting the above hypothesis. The schematic illustration of modulating effects of caffeine on blood platelet function is presented in the Figure 2.

## 6. Adenosine Receptor-Mediated Effects of Caffeine on the Cardiovascular System

Adenosine plays an important role as a regulator and modulator of the cardiovascular system [102]. It exerts its cardiovascular effects mainly through A_1_ adenosine receptors [73], a subtype involved in the regulation of blood pressure [103]. Caffeine blocks adenosine receptors, and a complete blockade increases systemic levels of adenosine [104]; thus, circulating chemoreceptors and other receptors are stimulated, increasing sympathetic tone, catecholamine levels, peripheral vascular resistance, and renin secretion [104]. Stimulation of the A_1_R leads to a contraction of vascular smooth muscle and to a decrease in coronary blood flow [105]. In addition, caffeine alters total peripheral resistance, diuresis and heart rate [72].

Caffeine can have a dual effect on blood pressure by antagonising the adenosine receptors A_1_R, A_2A_R and A_2B_R [72]. While caffeine was historically believed to increase blood pressure, recent studies suggest a different relationship between adenosine and caffeine. Although blood pressure indeed increases shortly after caffeine consumption, it does not seem to have long-term effects [106]. Studies indicate that while caffeine has an acute effect on both systolic and diastolic blood pressure, long-term consumption does not appear to have a chronic effect [72,104,107]. Interestingly, habitual coffee drinking may in fact reduce the risk of cardiovascular disease and hypertension [106]. Caffeine has been shown not only to affect blood pressure but also to increase systemic vascular resistance [108].

The pressor effect of caffeine acts mainly by altering vessel resistance [109]. Caffeine is also able to improve endothelial cell function by increasing cellular calcium, thus stimulating nitric oxide (NO) production [104]. NO stimulates vasodilation by acting on vascular smooth muscle [104,110]. Studies suggest that caffeine affects arterial stiffness in both normal and hypertensive subjects but that this effect lasts longer in the second group [111,112]; this hypothesis has been supported by other studies [113,114]. In addition, chronic coffee consumption also has a negative effect on wave reflections in healthy subjects [113].

However, regular coffee consumption has positive effects. The habit of drinking coffee may be associated with a lower risk of developing hypertension, heart failure, and atrial fibrillation [115]. Studies suggest the existence of a J-shaped relationship between caffeine consumption and the risk of developing coronary heart disease [115]; i.e., moderate caffeine consumption is associated with a reduced risk of CVD and may even be protective against CVD, whereas heavy coffee consumption is associated with an increased risk of CVD [115,116].

## 7. Concluding Remarks

Caffeine is known to affect various aspects of physiology and may stimulate or weaken the activity of some of the drugs. Although for many years it was thought to have a negative effect on blood pressure, cardiovascular events, and morbidity, recent evidence suggests the opposite. Caffeine consumption appears to have a range of health benefits, including a protective effect against CVD or neurodegenerative diseases. As it was shown in many recent observational studies, coffee and tea drinking significantly reduced cardiovascular risk and mortality. It is worth noting that the effects of caffeine on the human body, beneficial or negative, depend on the dose taken and the frequency of consumption. 

In the human body, caffeine acts on multiplate targets. The most important mode of action for caffeine is a blockade of adenosine receptors which plays a principial role in mediating its effects on the cardiovascular system and blood platelets. Adenosine receptors play a crucial role in blood flow and pressure regulation, vasodilation, heart rate regulation, and cardioprotection, and their activation with adenosine or adenosine agonists causes the inhibition of platelet function. Also, adenosine agonists, especially for A_2A_AR and A_2B_AR, are being considered as enhancing substances for antiplatelet therapy, e.g., with P2Y_12_ inhibitors; hence, this study is being conducted to synthesize new products that would exhibit selective activity against platelets. Caffeine is known as a blood platelet modulator; although dependent on dose and term of caffeine use, caffeine intake may stimulate or inhibit platelet reactivity. In addition, chronic intake of coffee alters the response of platelets to the activity of adenosine- or NECA-induced (NECA—non-selective agonist of adenosine receptor) platelet inhibition, and intake of caffeine for two weeks may upregulate A_2A_ adenosine receptor activity, increase cAMP accumulation and anti-aggregatory effects, and decrease calcium levels elicited by HE-NECA. These results from several studies indicate that chronic exposure to caffeine may sensitize or upregulate the adenosine receptors in platelets. However, further research is needed on this topic, focused on the one hand on the search for selective agonists for adenosine receptors with antiplatelet effects and, on the other hand, on the role and mechanisms of caffeine’s action on platelets and the circulatory system 

Overall, consumption of coffee, the richest source of dietary caffeine, seems to be beneficial. Moderate caffeine intake may be associated with a lower risk of CVD. Its consumption also has anti-thrombotic and antiplatelet effects and appears to inhibit platelet aggregation. Knowledge of the relationship between coffee consumption and platelet aggregation may be of great value for both nutritionists and clinicians: not only because caffeine may affect human health but also because it may be able to modulate the effects of some drugs, especially those used in antiplatelet therapy. 

## Figures and Tables

**Figure 1 ijms-25-08905-f001:**
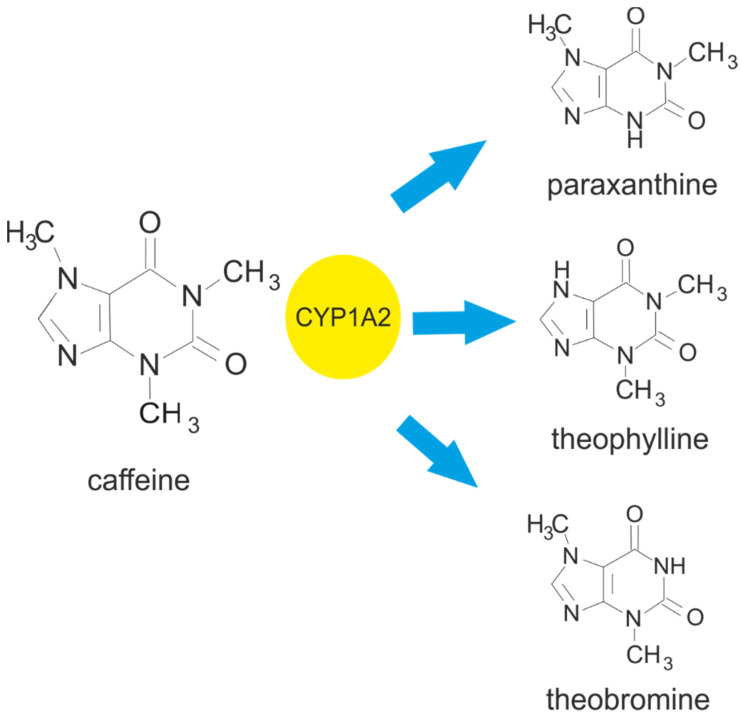
Metabolism of caffeine leading to the formation of the main metabolites: paraxanthine, theophylline, and theobromine. Cytochrome P450 isoform 1A2 (CYP1A2) is responsible for 3-demethylation of the caffeine, leading to the formation of 1,7-dimethylxanthine (paraxanthine). It also facilitates the 1- and 7-demethylation of caffeine, which results in the formation of 3,7-dimethylxanthine (theobromine) and 1,3-dimethylxanthine (theophylline).

**Figure 2 ijms-25-08905-f002:**
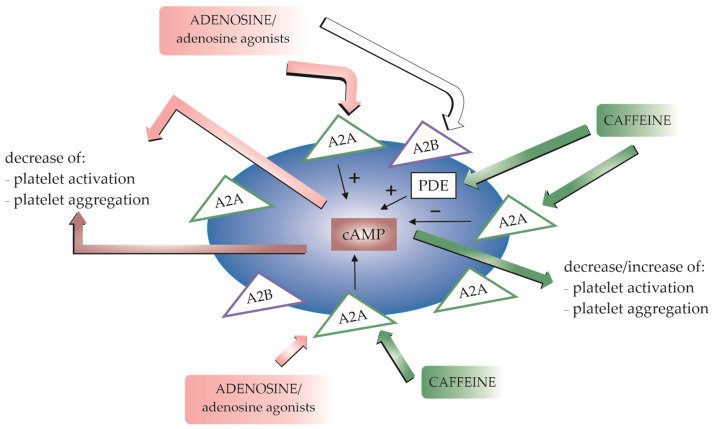
A schematic illustration of modulating effects of caffeine on blood platelet function. The adenosine receptors A_2A_ and A_2B_ are expressed in platelets. A_2A_ receptors are predominantly involved in adenosine-stimulated platelet activation. Stimulation of adenosine receptors increases 3′,5′-cyclic adenosine monophosphate (cAMP) production, resulting in a decrease in platelet activation and aggregation. Caffeine affects platelet reactivity by antagonising adenosine receptors. The antiplatelet effect of caffeine can be enhanced by the presence of adenosine or other agonists. Caffeine may also exert its antiplatelet activity due to the inhibition of phosphodiesterase (PDE) activity, resulting in an increase in cAMP levels.

**Table 1 ijms-25-08905-t001:** Summary of biological and adverse effects of caffeine consumption.

Biological Effects	Adverse Effects
Modification of lipid and glucose metabolism [32,33]	Acute intoxication [10]
Enhance of hormonal secretion [34]	Tachycardia [10]
Elevation of neurotransmitters levels [34]	Arrhythmia [10]
Reduction of inflammatory biomarkers [35]	Anxiety [40,41]
Activation of anti-inflammatory mechanisms [10,36]	Decrease in sleep duration [31]
Enhance of contractility [16]	Addiction [42]

**Table 2 ijms-25-08905-t002:** Role of adenosine receptor (AR) subtypes in cardiovascular system.

Subtype of Adenosine Receptors	Role in Cardiovascular System	Citation
A_1_	Vessel tone regulation	[69]
	Heart rate reduction	[56]
	New vessel formation	[56]
	Cardioprotection	[70]
A_2A_	Vasodilation	[56]
	Wound healing	[56]
	Angiogenesis	[71]
	Vasculogenesis	[71]
	Blood pressure regulation	[72]
	Blood flow regulation	[73]
	Cardioprotection	[70]
	Inhibition of platelet function	[74]
A_2B_	Vasodilation	[56]
	Blood pressure regulation	[72]
	Blood flow regulation	[73]
	Angiogenesis	[71]
	Vasculogenesis	[71]
A_3_	Cardioprotection	[70]
	Blood pressure regulation	[72]

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
