# Peer review of "The Effects of Caffeine on Blood Platelets and the Cardiovascular System through Adenosine Receptors"

_ijms, 2024, doi:10.3390/ijms25168905_

Round 1
Reviewer 1 Report
Comments and Suggestions for Authors
In general the review article is well-written, with detailed introduction on caffeine, adenosine receptors and the effect of platelet on cardiovascular systems.
I just have one suggestion, for the part discussed the role of caffeine mediated by adenosine receptors on platelet or cardiovascular system, it will be easier for the reader to figure our the logic behind the words if the author could add a figure indicating the mechanisms.
Thank you!
Author Response
Please see the attachement.

Reviewer 2 Report
Comments and Suggestions for Authors
A review article by Marcinek et al. summarizes the possible effects of caffeine on blood platelets and the cardiovascular system through adenosine receptors. The article is well-written. Data are presented progressively, from describing the nature of caffeine and its general biological properties to the mechanisms by which caffeine exerts its effects on the cardiovascular system. The adenosine receptors were defined as key players in mediating caffeine effect on cells through inducing intracellular changes. However, this article still needs several improvements:
The methodology of the work is missing. The authors should explain the searched databases and the used keywords.
Abbreviations should be consistent throughout the article. Whenever possible, write the full names followed by the abbreviations in parentheses. Lines 175, 285 should be corrected accordingly.
Regarding the topic of this work, the use of caffeine as an adjuvant to plate inhibitors is well documented (PMID: 17892993). However, the efficacy of its standalone use is still doubtful. Coffee contains other phenolic compounds besides caffeine. Positive effect of these compounds on platelet inhibition has been suggested, as coffee drinking inhibited the aggregation of platelets induced by several agonists, while caffeine alone failed to achieve similar effects. (PMID: 18439332)
Diagrams summarizing the biological roles of caffeine on the cells of the cardiovascular system as well as the key playing factors (e.g., adenosine receptors, cAMP, etc.) will help to improve the readability of this work.
The concluding section, especially parts related to future directions, requires significant improvements.
Comments on the Quality of English LanguageMinor edits to the English language are needed.
Author Response
Please see the attachement.

Reviewer 3 Report
Comments and Suggestions for Authors
The manuscript presented by Marcinek and co-workers, entitled „ Effects of caffeine on blood platelets and the cardiovascular system through adenosine receptors” (ijms-3098486) is a review presenting literature data on the topic specified in the work's title.
The presented manuscript is interesting, but some changes must be made before publication.
Please find below my major comments. 
- the work does not clearly demonstrate the innovative nature of the research described
- The abstract does not present the scientific importance of the study, 
- the work lacks the classification of caffeine metabolites (phase I and II). A longer discussion on the metabolites should be presented.
- the suitable solution will be to collect all information, including the biological effects of caffeine and its adverse effects, in the table. The same is in the case of adenosine receptors-mediated effects of caffeine on platelets.
- the future prospects in the study of caffeine effects should be presented.
- The presented literature does not fully present the latest scientific achievements in the presented topic. The authors also did not present any significant review works on this topic from recent years. This needs to be supplemented.
Comments on the Quality of English Language
Moderate English revision is required.
Author Response
Please see the attachement.

Reviewer 4 Report
Comments and Suggestions for Authors
The manuscript reviews the impact of caffeine on the cardiovascular system, particularly its effects on blood platelets, suggesting that while caffeine was historically viewed as detrimental due to its blood pressure-raising effects, recent evidence indicates it may reduce the risk of cardiovascular diseases and hypertension. Additionally, it explores the potential of caffeine's antiplatelet activity and discusses the concept of dual antiplatelet therapy, proposing the combination of P2Y12 antagonists and adenosine receptor agonists as a novel cardioprotective strategy.
The manuscript is valuable, addressing a significant topic in recent years. In the abstract, I recommend indicating the basis for caffeine's antiplatelet activity.
The article does not include important references, such as: https://www.sciencedirect.com/science/article/pii/S1043661822005424?via%3Dihub
The introduction should specify what has been published so far and what is novel in this manuscript.
In subsection 2.4, the biochemical basis for the adverse effects and toxicity of caffeine should be indicated.
The sentences in subsection 3.2 should be linked to caffeine instead of generally describing the function of adenosine receptors.
For chapter 5, I recommend creating a biochemical graphic illustrating what is described in it. This will enhance the clarity of the manuscript.
Comments on the Quality of English LanguageModerate editing of English language required.
Author Response
Please see the attachement.

Round 2
Reviewer 2 Report
Comments and Suggestions for Authors
Authors have satisfactorily addressed my concerns. The manuscript is improved and might be considered for publication.
A few typos were detected in the revised version to be further considered:
Line 18: dependently on the dose and term
Line 62, 63: depending on the dose
Line 73: March 31, 2023
Line 101: is mediated by
Line 186: activation of β1-receptor
Line 188: intensify the production
Line 190: In the brain,
Table 1. 2nd and last row, Enhance of, delete of
Comments on the Quality of English LanguageThe English of the manuscript is of accepted quality.
Reviewer 4 Report
Comments and Suggestions for Authors
The recommended changes have been implemented.
Comments on the Quality of English LanguageMinor editing of English language required.